# Thirty-Nine-Year Wave Hindcast, Storm Activity, and Probability Analysis of Storm Waves in the Kara Sea, Russia

Stanislav Myslenkov [1,2,3,*] , Vladimir Platonov [1] , Alexander Kislov [1] , Ksenia Silvestrova [2] and Igor Medvedev [2,4]

[1]  Department of Oceanology, Lomonosov Moscow State University, 119991 Moscow, Russia; vplatonov86@gmail.com (V.P.); avkislov@mail.ru (A.K.)
[2]  Shirshov Institute of Oceanology RAS, 117997 Moscow, Russia; ksberry@mail.ru (K.S.); medvedev@ocean.ru (I.M.)
[3]  Hydrometeorological Research Centre of the Russian Federation, 123242 Moscow, Russia
[4]  Fedorov Institute of Applied Geophysics, 129128 Moscow, Russia
[*]  Correspondence: stasocean@gmail.com

**Abstract:** The recurrence of extreme wind waves in the Kara Sea strongly influences the Arctic climate change. The period 2000–2010 is characterized by significant climate warming, a reduction of the sea ice in the Arctic. The main motivation of this research to assess the impact of climate change on storm activity over the past 39 years in the Kara Sea. The paper presents the analysis of wave climate and storm activity in the Kara Sea based on the results of numerical modeling. A wave model WAVEWATCH III is used to reconstruct wind wave fields for the period from 1979 to 2017. The maximum significant wave height (SWH) for the whole period amounts to 9.9 m. The average long-term SWH for the ice-free period does not exceed 1.3 m. A significant linear trend shows an increase in the storm wave frequency for the period from 1979 to 2017. It is shown that trends in the storm activity of the Kara Sea are primarily regulated by the ice. Analysis of the extreme storm events showed that the Pareto distribution is in the best agreement with the data. However, the extreme events with an SWH more than 6-7 m deviate from the Pareto distribution.

**Keywords:** the Kara Sea; wave climate; storm activity; wind waves; wave modeling; WAVEWATCH III; probabilistic analysis; extreme waves

## 1. Introduction

The interest increases in the study of the hydrometeorological conditions of the Arctic seas due to the active economic development of this region. The active oil and mineral field exploration and development occur here, in this region. The Arctic is an area of intensive shipping and fishery. Wind, sea ice, and wave conditions are limiting factors for the economic activity and the development of the infrastructure in the coastal zone. The storm waves can destruct the infrastructure facilities in coastal zones and offshore, that threaten human life and cause economic damage.

We need to study the extreme winds and waves in the past, their interannual variability because it is possible to reduce the disasters' risk in the future.

Nowadays, storm activity is studied with several methods with the use of different sources: Direct observation data [1,2], altimetry data [3–5], and modeling data [6–11]. There is also research work on wave heights in the 21st century Arctic Ocean [12]. As direct measurements, especially in the Arctic Region, are very rare, and altimetry data are short series, thus the simulated data from models are more suitable for spatio–temporal analysis. Statistical analysis of observation and modeling data often used for investigations of long-term variability of extreme wave height [1,2,13,14].

Below we will consider the publication which most closely related to the wave climate and storm activity in the Kara sea and in the whole Arctic region.

Regular and extreme characteristics of wind and waves of the Kara Sea are given in the Wind and Wave Climate Handbook of the Russian Maritime Register of Shipping [15]. These data are based on the results of modeling. But the input wind forcing for the simulations was calculated from the atmospheric pressure data. Subsequently, it was verified and calibrated by the weather stations measurements. Such information needs to be refined with the modern atmospheric reanalysis data. Diansky et al. (2014) describe some new results devoted to wave hindcast and forecast of the Kara Sea using the WRF wind [16].

Stopa et al. (2016) showed the main features of the wave climate and trends in the whole Arctic for the period 1992–2014 based on altimetry data and wave hindcast results [17]. They noted that the ice cover decreases and simultaneously the wave height rises. Liu et al. (2016) used satellite observations (1996–2015) for studying the wave climatology in the Arctic Ocean in summer (August–September) [5]. They show that winds and waves in the Barents and Kara Seas initially increased from 1996 to 2006 and later decreased until 2015.

Li et al. (2019) present details of the significant wave height (SWH) change with the retreat of the ice edge [18]. The increase of the wave heights is shown for the Arctic subregions, including the Kara Sea.

Interannual variations of the mean and extreme SWH in ice-free conditions in the Kara Sea are described in [19]. The estimated linear trends of SWH from 2005 to 2018 provided, but these trends are not statistically significant for most areas. The mean and extreme SWHs show relatively positive trends in the northeastern part of the Kara Sea, but the analyzed period is too short for trend estimation [19].

Positive trends of the highest SWH and wind speed are shown for the Laptev and the Beaufort Seas based on the 38-year-long reanalysis. But for the Kara Sea trend analysis was not realized [20].

The wind wave characteristics are studied in several researches for the whole world ocean [8,9,21]. Semedo et al. (2011) described the seasonal variations of the global wave heights from 1957 to 2002 with the ERA-40 reanalysis data [9]. In the Barents Sea, the positive linear trend of SWH in winter months is observed by Semedo et al. (2011).

It is also important to consider several works devoted to the analysis of ice cover, meteorological, and oceanic characteristics in the Arctic ocean, which can take influence in storm activity.

In [22] studied the seasonal and interannual features of the Kara Sea meteorological regime and its connection with circulation indices. The period 2000–2010 is characterized by significant climate warming, a reduction of the surface of the old and first-year sea ice in the Arctic [23–25], and the appearance of a significantly larger ice-free sea surface than earlier. The influence of sea ice must certainly have a strong influence on the propagation of wind waves, it is indicated in the work [16]. Thermobaric structure of the atmosphere [26,27], and Atlantic water inflow [28] changed significantly over the last 40 years. These factors can lead to changes in the wind–wave regime in the Arctic Region. Such important features as a modification of the cyclone number and its trajectories [29,30], and the increase in daily extremes of wind speed [31] were described. Also, wind speeds rose to the north from 75–80° N in recent decades according to climatic reports [32]. Positive trends in average and extreme wind speeds in some parts of the Arctic Region are also noted in [4].

However, in all mentioned studies about the wave climate of the whole Arctic or the Kara Sea, there is no deep analysis of the storm interannual variability in the Kara Sea, or the data series are too short to carry it out. The main motivation of this research is to assess the impact of climate change on storm activity over the past 39 years in the Kara Sea.

In this research, we present the wave hindcast of the Kara Sea with a high spatial and temporal resolution. The regular and extreme wave characteristics were studied. The recurrence, trends, and probability analysis of the storm waves in the Kara Sea were estimated for the long period from 1979 to 2017.

## 2. Materials and Methods

### 2.1. Wave Modeling

One of the main approaches of studying the world ocean wave climate is the spectral wave modeling that allows creating long-term reanalyses of wave parameters [8,9,11,33].

Modern spectral wave models provide high-quality results, which are in good agreement with direct wave measurements. Correlation between model results and measurements data is usually 0.8–0.9, and the standard RMSE error is 0.3 m [17,18,33].

The wave characteristics in the Kara Sea were calculated by the spectral wave model WAVEWATCH III (WWIII). We use two versions of WWIII 4.18 [34] and 6.07 [35]. This model considers such parameters as: Wind speed, ice concentration, effects of the energy dissipation, non-linear interactions, and bottom friction. This model is based on a numerical solution of the equation of the wave action density spectrum:

$$\frac{DN}{Dt} = \frac{S}{\sigma} \tag{1}$$

where, $N(k, \theta) \equiv F(k, \theta)/\sigma$, $k$—wavenumber, $\theta$—propagation direction, $\sigma$—relative frequency, $D/Dt$ represents the total derivative (moving with a wave component), and $S$ represents the net effect of sources and sinks for the spectrum $F$ [34]. $S$ is a function that describes the transfer of the energy from the wind to the waves, nonlinear wave interactions, dissipation of the energy through the collapse of the crests at a great depth and in the coastal zone, friction against the bottom and ice, wave scattering by ground relief forms, and reflection from the coastline and floating objects. The energy balance equation is integrated using finite-difference schemes by the geographic grid and the spectrum of wave parameters.

In the present study, the calculations were made by using the model version 4.18 with ST1 scheme [36] and Komen et al. (1984) [37,38] and the model version 6.07 with ST6 scheme [39,40]. We chose this because at first we used the old version of the model and the sensitivity tests showed good quality of ST1 scheme. A new version of the model was released in 2019 and we decided to evaluate it quality too.

Other settings were the same for both wave model realizations. A Discrete Interaction Approximation (DIA) model was used for the possible nonlinear interactions of the waves [41]. The DIA is a standard approximation for the calculation of nonlinear interactions in all modern wave models. The influence of the sea ice on the wave development was considered by the IC0 scheme, where a grid point is considered as ice-covered if ice concentration is >0.5.

In the shallow water, the increase in wave height as waves approach the shore, and the related wave breaking after waves reach the critical value of steepness were taken into consideration. The standard JONSWAP scheme was used to take the bottom friction into account [42]. The spectral resolution of the model is 36 directions (Dq = 10°), the frequency range includes 36 intervals (from 0.03 to 0.843 Hz). The total time increment for the integration of the complete equation of the wave action is 15 min.

The calculations were performed using the unstructured grid, which consists of 37729 nodes (Figure 1). The bathymetric data were obtained from the ETOPO 1-minute bottom topography database (https://www.ngdc.noaa.gov/mgg/global/, accessed on 27 February 2020) and detailed navigation maps. The grid covers the Barents and Kara seas, as well as the entire northern part of the Atlantic Ocean. The spatial resolution varies from ~700 m for the coastal zone of the Kara Sea to ~50 km for the northern part of the Atlantic Ocean. The North Atlantic was included in the grid because of the swell propagating into the Barents and Kara seas, it was shown earlier in [43].

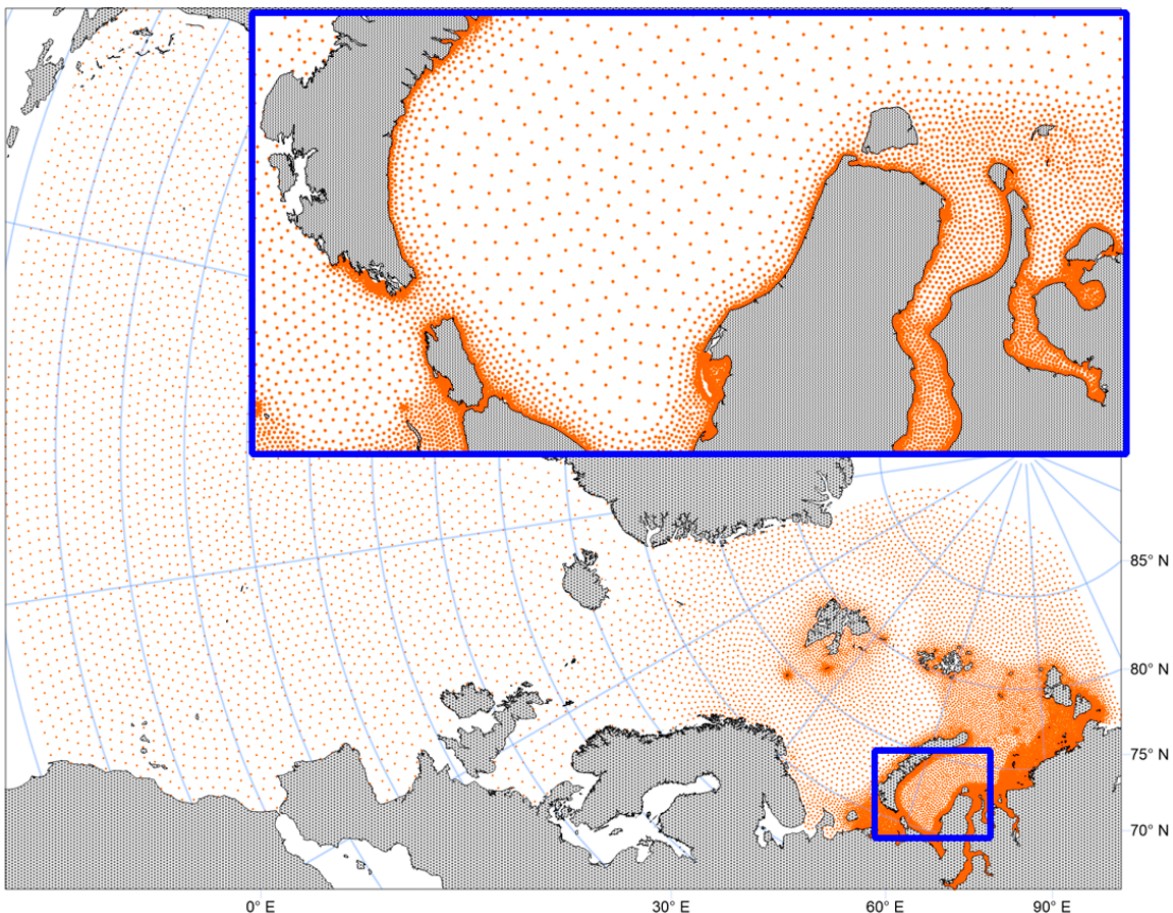

**Figure 1.** Unstructured computational grid of the WWIII model for the North Atlantic and the Kara Sea.

Wind speed on a 10 m above the ground and sea ice concentration data for the wave modeling were taken from the NCEP/CFSR reanalysis (1979–2010) with a spatial resolution ~ 0.3° [44] and NCEP/CFSv2 reanalysis (2011–2017) with a resolution ~ 0.2° [45], and temporal resolution of 1 hour. We get the reanalysis data from server https://rda.ucar.edu (accessed on 27 February 2020). Linear interpolation of reanalysis data to unstructured mesh performed by using own program code. Further, in the wave model, the interpolated wind was used in the mode "as is".

A more detailed description of the model configuration, the main features of the experiments with the unstructured mesh is presented in [46–48].

As a model output, we got the wind wave fields for every three hours from 1979 to 2017 (total 39 years). We tested the data with a time step of 1 hour and 3 hours and did not reveal a significant change in the extremes (no more than 0.1 m). The model results include the SWH ($4\sqrt{m_0}$, where $m_0$ is the zero-order moment of the wave spectrum, approximately SWH is the mean value from 1/3 of the highest waves), the wave propagation direction, the mean wave period (WP) Tm02=($2\pi\sqrt{\overline{\sigma^2}}$), and mean wavelength (WL)= ($2\pi\overline{k^{-1}}$). Also, the wave heights of 1% and 3% probability of exceedance (it mean that 1% of the single waves are higher than 99% other waves during the 15 min period) were used for the data analysis. These values were calculated as 1.51 × SWH and 1.32 × SWH, respectively [49,50]. SWH and wave height with other probability calculated in the model for 15 min integration interval. The maximum and long-term SWH were calculated based on these data. When the Kara Sea was ice covered the wave parameters were equal to zero in model results. The mean long-term characteristics were performed for the ice-free period when the wave parameters were nonzero.

### 2.2. Quality Assessment of the Wave Model Results

The instrumental wave measurement data were used for the quality assessments of the modeling results. Wave measurement data collected at a mooring station in the Kara Sea (Figure 2) for the September–October 2012. Wind and wave parameters were measured with upward-looking sonar IPS-5 for ice profiling, and were published in [51]. Authors digitized graphs from the Atlas for statistical analysis.

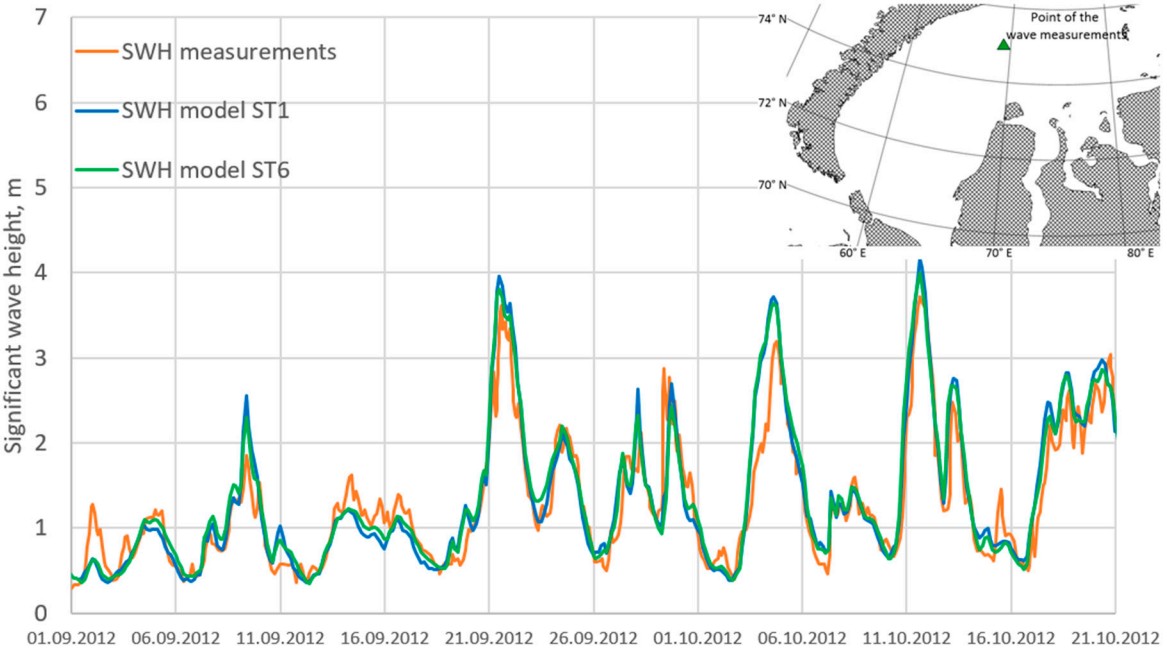

**Figure 2.** The measured and simulated significant wave height (SWH) for mooring station in the Kara Sea, location of the wave measurement station marked on insert map.

The model quality assessments based on the standard statistical parameters:

$$\text{Bias} = \sum_{i=1}^{N} \frac{1}{N}(P_i - O_i) \tag{2}$$

$$\text{RMSE} = \sqrt{\frac{1}{N-1}\sum_{i=1}^{N}(P_i - O_i)^2} \tag{3}$$

$$\text{SI} = \frac{\text{RMSE}}{\frac{1}{N}\sum_{i=1}^{N} O_i} \tag{4}$$

$$R = \frac{\sum_{i=1}^{N}\left((P_i - \overline{P})(O_i - \overline{O})\right)}{\sqrt{\left(\sum_{i=1}^{N}(P_i - \overline{P})^2\right)\left(\sum_{i=1}^{N}(O_i - \overline{O})^2\right)}} \tag{5}$$

where, N—is the total number of data, $P_i$—is the model value, $O_i$—is the observed value, $\overline{P}$—is the mean model value, $\overline{O}$—is the mean observed value.

A comparison of the modeled and measured SWH from 1 September to 22 October, 2012 for the mooring station is shown in Figure 2. The both variants of model calculations provides the absolute wave height and the phase of the individual storm event quite well. The result of the comparison as a scatter diagram is shown in Figure 3. The R (correlation coefficient) is 0.94, the BIAS is 0.08 m, and the RMSE is 0.31 m. The Scatter Index is 0.28. Overestimation (0.2–0.4 m) of the model SWH is observed for several storms. Further in

the analysis, the SWH values are presented with an accuracy of one decimal place due to the obtained quality estimates.

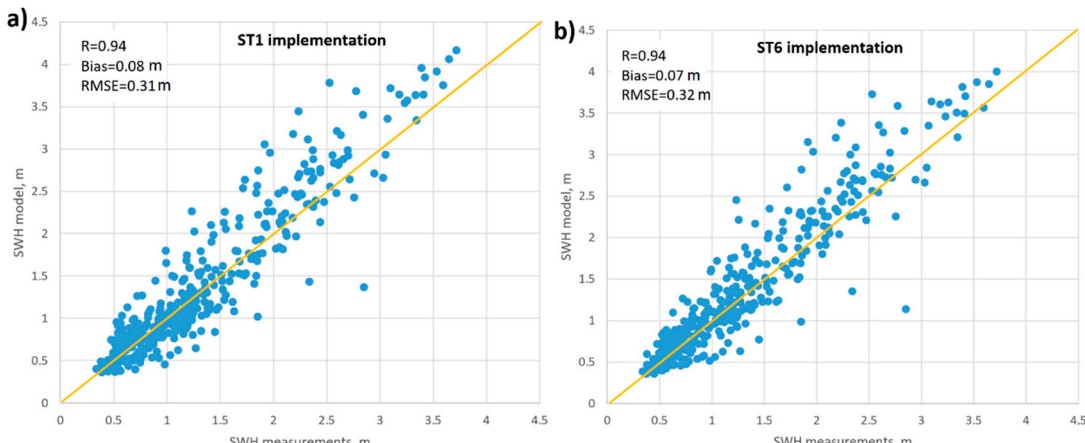

**Figure 3.** Scatter diagram of measured on the mooring station and simulated SWH for two model implementations: (**a**) ST1, (**b**) ST6.

Because the duration of direct measurements is too short, we decided to use IMOS satellite data for quality assessments. The satellite database contains global data of wind speed and wave height obtained from all the altimeter missions since 1985. The data has been calibrated against NDBC buoy data and validated with independent buoy measurements and at cross-over points with other altimeter missions [52]. We used three modern altimetry missions—Cryosat, Saral, and Sentinel, which have a better quality of SWH measurements against old missions Topex or Jasons 1,2. The part of satellite data was filtered (deleted) from the analysis when it had bad quality flags or was closer than 10 km to the shore or ice edge. The distance between the pairs of compared points from the models and the satellite was no more than 13 km. More than 190,000 points of satellite SWH were collected for the area of the Kara Sea.

The results of quality assessments based on the satellite data of Cryosat, Saral, and Sentinel are provided in Table 1 and Figure 4. Basic statistical parameters are presented in Table 1. The results are similar to assessments, which based on direct measurements. We obtained the model overestimation (~0.1 m) for ST6 implementation, shown on Figure 4. The obtained results allow us to suggest that the quality of both implementation is approximately the same. Further, we decided to use in wave analysis the results from ST1 implementation, because its BIAS error was smaller than in ST6.

**Table 1.** Basic statistical parameters of model quality assessments.

| Sat/Parameter | R | Bias, m | RMSE, m | SI | N | Years |
|---|---|---|---|---|---|---|
| | | | **ST1** | | | |
| **Cryosat** | 0.89 | −0.07 | 0.39 | 0.3 | ~83,000 | 2010–2017 |
| **Saral** | 0.92 | 0.05 | 0.32 | 0.24 | ~74,000 | 2013–2017 |
| **Sentinel** | 0.91 | 0.07 | 0.37 | 0.27 | ~34,000 | 2016–2017 |
| | | | **ST6** | | | |
| **Cryosat** | 0.89 | −0.03 | 0.38 | 0.28 | ~83,000 | 2010–2017 |
| **Saral** | 0.93 | 0.11 | 0.33 | 0.24 | ~74,000 | 2013–2017 |
| **Sentinel** | 0.92 | 0.14 | 0.37 | 0.26 | ~34,000 | 2016–2017 |

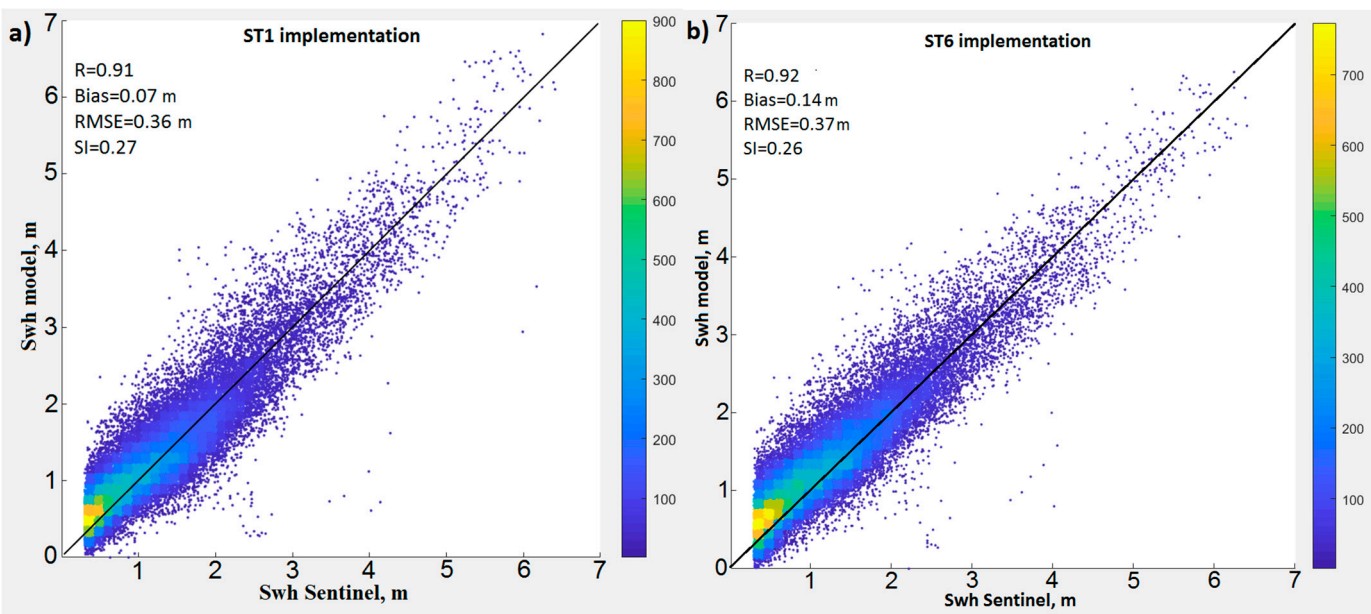

**Figure 4.** Scatter diagram of simulated SWH and Sentinel data for two model implementation: (**a**) ST1, (**b**) ST6.

The obtained quality assessments coincide with the other modern wave model implementations [17,18,33]. The quality of the modeled data allows estimating the regular and extreme characteristics of the wave climate, as well as the interannual variability of storm activity. Our model implementation may not be very accurate in absolute wave heights, scatter index = ~0.27. However, the main focus of this article is the interannual variability of storms and their climatic trends, which depend a little on absolute errors.

### 2.3. Recurrence of the Storm Wave Events

The storm activity analysis was held according to the Peak Over Threshold (POT) method, which is widely used [1,2,50]. The essence of the POT method is to find an extreme values of some sample that exceeds a certain threshold value. We used POT previously for the Barents Sea wave analysis [46]. The number of storm waves with different SWH from 3 to 7 m was calculated for each year in the whole Kara Sea or in the sectors of the sea (the description of allocation into sectors is given in Section 3.3). The calculation procedure included the following steps: If at least one node in the investigated sea area had the SWH exceeding for example a 3 m (or different threshold from 3 to 7 m), then such event was attributed to the storm case with SWH threshold 3 m. This event continued until the SWH was not less than the threshold at all nodes of the investigated area. Further, if the SWH threshold exceeded in one of the nodes again, then this event was added to the following case. A period of at least 9 hours passed between two storm cases for eliminating the possible errors. This technique has an inaccuracy associated with storms running in a row or from different directions at the same time. However, such cases are rare. The proposed algorithm works correctly; it was validated by a visual analysis conducted for several years.

The SWH 3 m is a 99% percentile of the whole data series (period 1979–2017, 3-hour interval), which was chosen as the lowest SWH threshold for the most part of the deep Kara Sea. For the ice-free period, SWH = 3 m is the 95% percentile. Therefore, further in the article, we will consider a deep analysis of the extreme events with the SWH exceeding 3 m for any points of the Kara Sea.

## 3. Wave Climate

### 3.1. Mean and Extreme Wave Parameters

The general features of the wave climate in the Kara Sea are discussed in this chapter.

The distribution of the maximum SWH and mean long-term SWH for the Kara Sea for the modeling period (1979–2017) is shown in Figure 5a–d. The mean long-term SWHs were about 1.1–1.3 m (Figure 5a) in the ice-free period. The maximum mean SWH was 1.3 m and observed in the northern part of the Kara sea. This area is associated with the influence of storms coming from the Barents Sea in the ice-free period. Formally the maximum SWH during the whole period reached 9.9 m and was observed in the northern part of the sea, at the border with the Barents Sea (Figure 5b). However, the wave conditions of this area were largely determined by the Barents Sea and this area belongs to the Kara Sea because of the formal border. In the central part of the Kara Sea, the SWH maximum is 9.4 m and it is observed off the western coast of the Yamal Peninsula (Figure 5b). The maximum wave height of a 3% probability is 12.4 m (Figure 5c), and a 1% probability is 14.3 m (Figure 5d) for the central part of the sea.

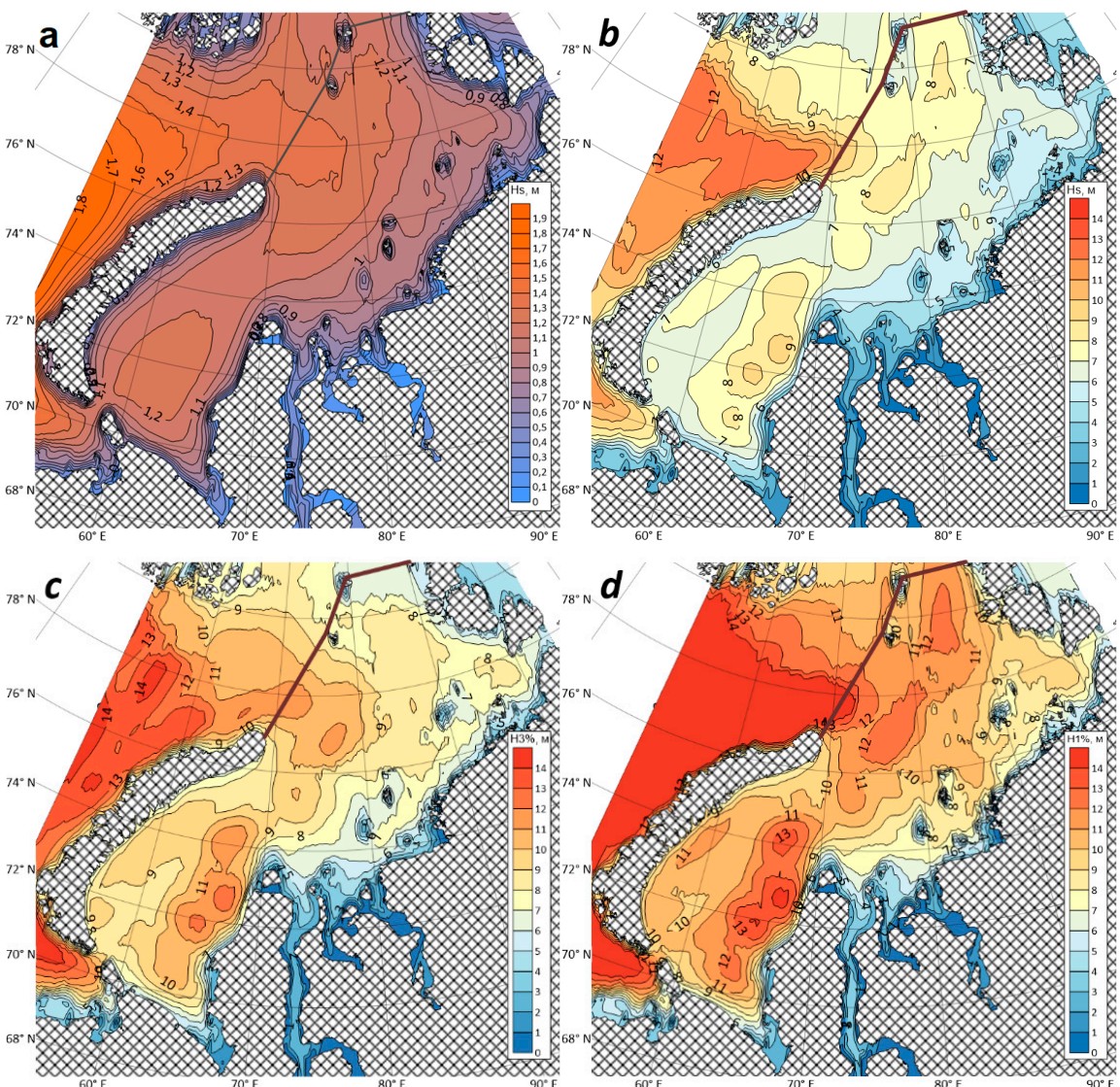

**Figure 5.** The long-term mean (**a**), maximum (**b**), significant wave heights, maximum wave height of 3% probability of exceedance (**c**), and maximum wave height of 1% probability of exceedance (**d**) according to the modeled data in the Kara Sea for the 1979–2017 period.

The Maritime Register Data [15] shows that in the Kara Sea the SWH with probability 1 time at 50 years is 5.4 m, and for a 1% probability of exceedance it is 7.8 m. Our results differ strongly from these estimates. It is explained by the model configurations and better wind forcing. Provided quality assessments for our wave model results allow the success of

this particular implementation. The ice period has been decreasing since 2009 in the Kara sea [24,53], so the number of extreme storms increases (see next section). Therefore, if the model calculates the period till 2006, results could differ significantly from the calculation period that ends in 2018.

A map of the long-term average probability of the ice occurrence is shown in Figure 6, obtained from the NCEP/CFSR/CFSv2 reanalysis data. This map is used for the analysis of the distribution of the maximum SWH and the distribution of the long-term mean SWH. In general, the maximum values of SWH and mean SWH are concentrated in the ice-free areas.

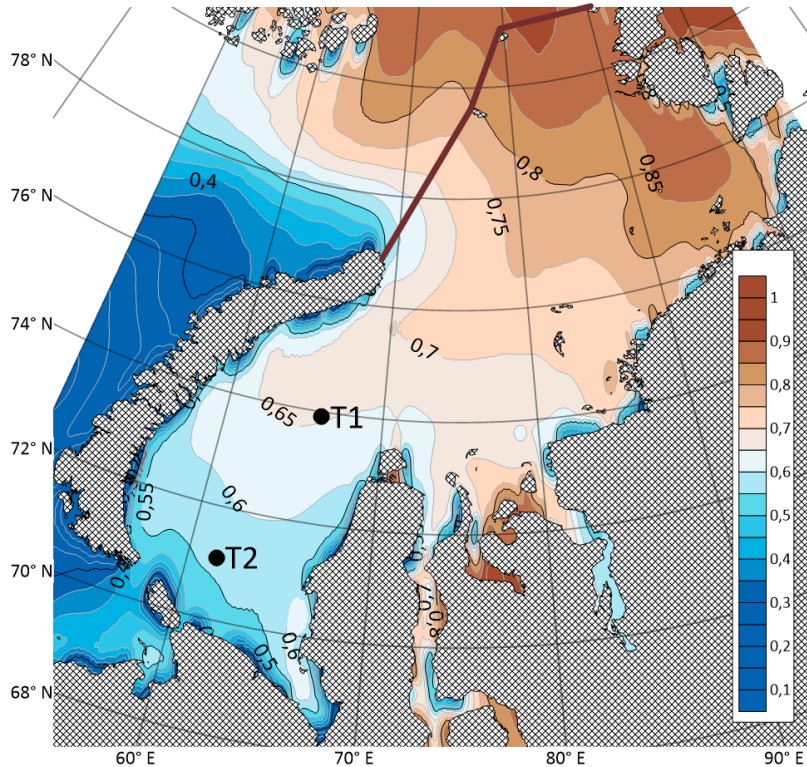

**Figure 6.** The long-term average probability of the ice presence of with a concentration more than 50% in the Kara Sea according to reanalysis data from 1979 to 2017 (in 0–1 unit). T1 and T2 are a points where the ice probability and wind events analyses provided.

According to long-term mean SWH fields and to maximum SWH values, at least we can reveal two large regions with particular spatio–temporal patterns of wave conditions, in the Kara Sea. The first one is the northern part, which is often occupied by the ice. It is affected by storms from the Barents Sea in the ice-free periods. The second region is the southwestern part of the sea (Figures 5 and 6). This region has a longer ice-free period and wave generation occurs without the influence of the Barents Sea. It should be pointed that in the north-eastern part of the Kara Sea the influence of storms from the Barents Sea should be expected, but, due to the high probability of the ice presence (>0.8) in this region, the wave height is significantly lower than in other parts of the sea.

The mean and maximum values of the average wave period and average wavelength are presented below. The long-term mean WP is 3.5 s (Figure 7a). Such small WP is due to the long ice period and as a consequence wave fetch is short. Mean WP corresponds to the mean long-term SWH of 1–1.3 m. The maximum WP is 8.4 s for the central Kara Sea and 10 s for the northern Kara Sea. Thelarge WP is caused by several storms that come from the Barents Sea, where fetch is significantly greater and swell has a longer period. The average wavelength is 30 m for the central Kara Sea and 35–40 m for the northern Kara Sea (Figure 7c). The maximum wavelength is from 160 m and up to 300 m (Figure 7d) for the central and northern Kara respectively. However, large values are due to the calculating

peculiarities of a wavelength in spectral models. The remnant swell with an insignificant wave height can provide a peak period of up to 20 s and a long wavelength in almost calm conditions, but these values should not be considered as extreme.

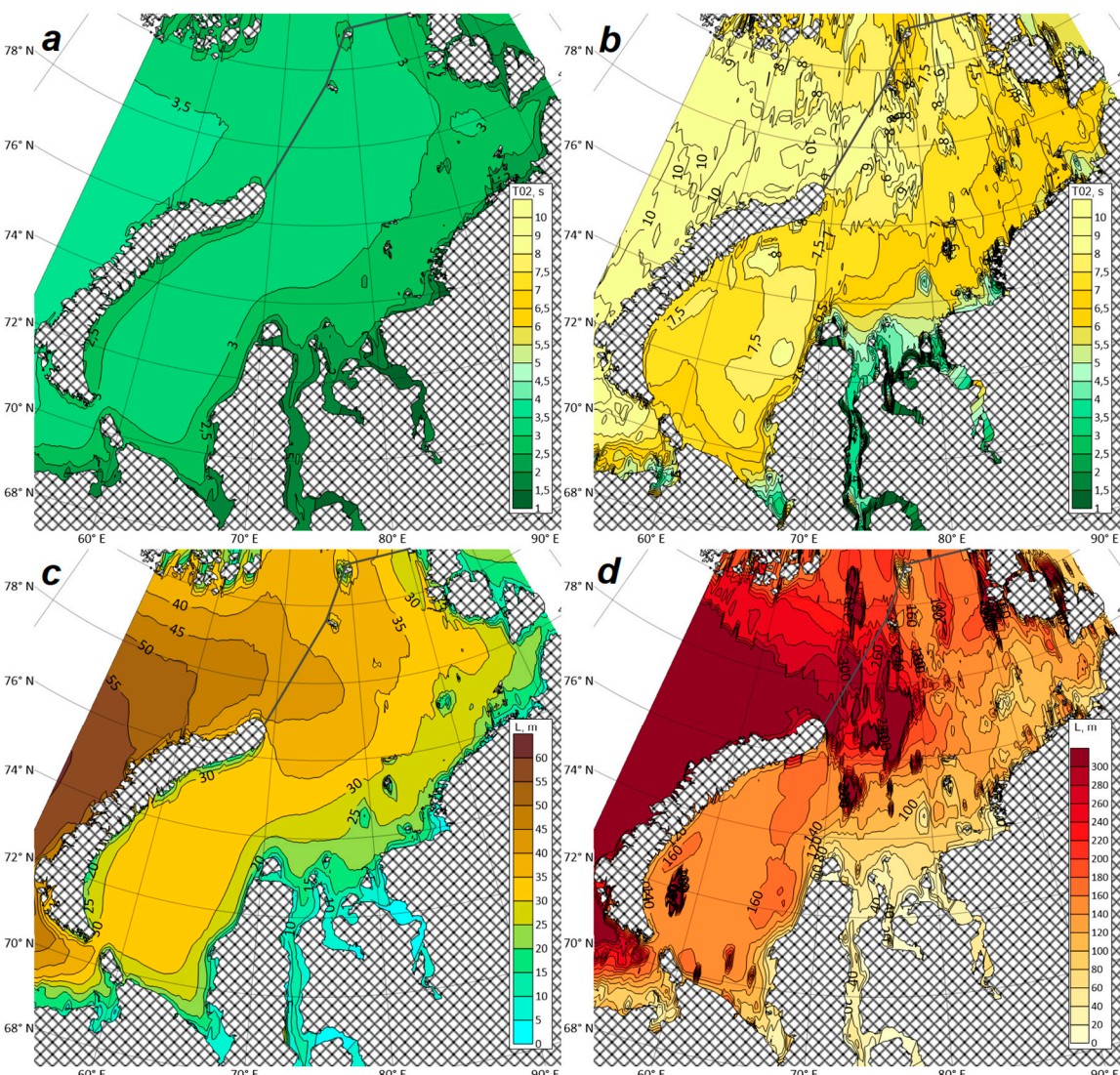

**Figure 7.** The long-term mean period (**a**), the maximum period (**b**), the long-term mean wavelength (**c**), and the maximum wavelength (**d**) in the Kara Sea according to modeling data for the period from 1979 to 2017.

### 3.2. Seasonal Variability of Wave Characteristics

The next step of our research was a seasonal analysis of the SWH maximum for four periods: December–January–February (DJF), March–April–May (MAM), June–July–August (JJA), and September–October–November (SON). Figure 8 shows the SWH maximum for different periods of the year for the entire simulated period. Seasonal SWH maxima variability is also influenced by the ice conditions of the Kara Sea. Probability maps of the ice presence (with a concentration of more than 50%) for the same seasonal periods according to reanalysis are shown in Figure 9.

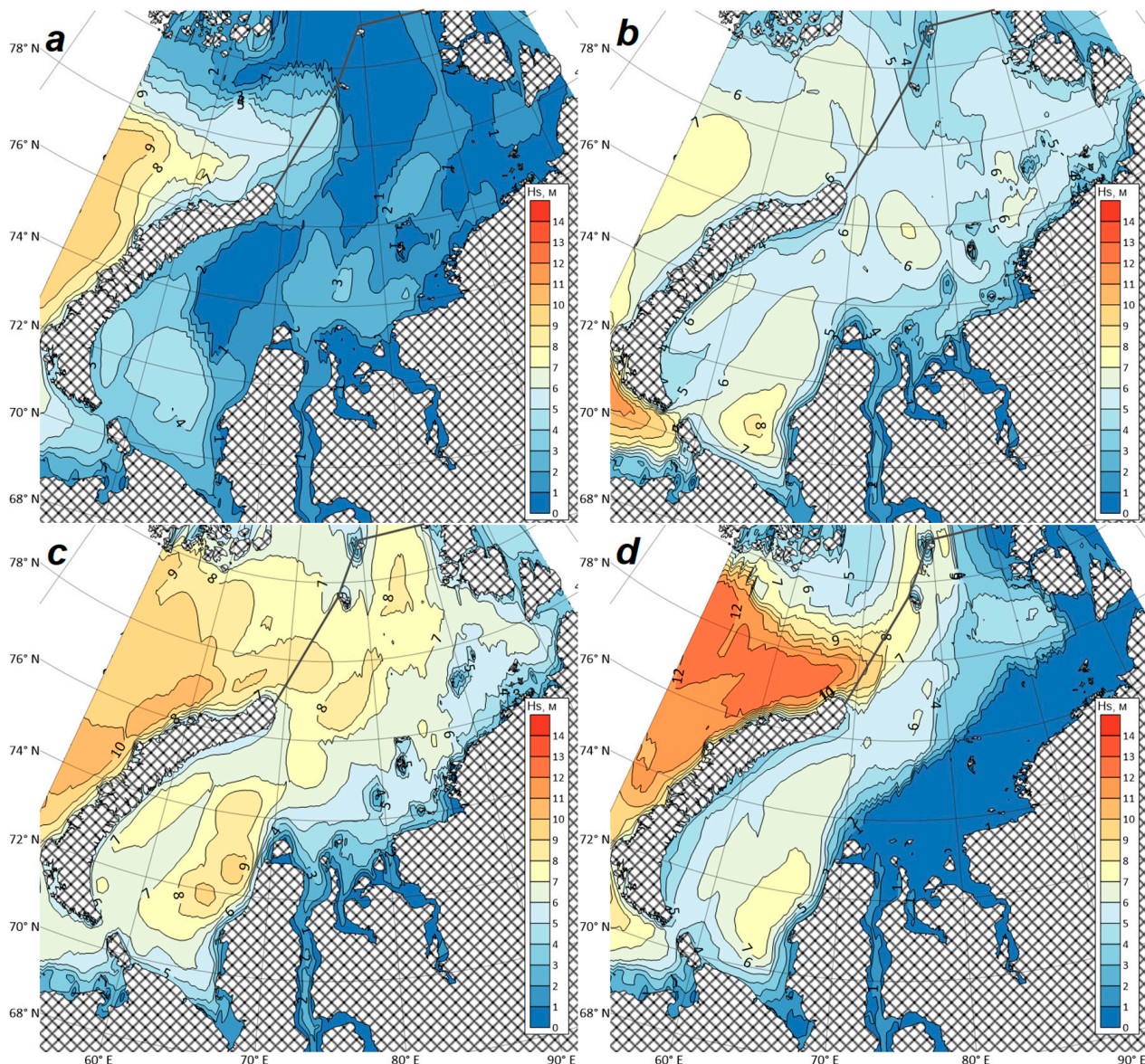

**Figure 8.** The maximum SWH in the Kara Sea according to the model data (from 1979 to 2017) for the periods: March–April–May (MAM) (**a**), June–July–August (JJA) (**b**), September–October–November (SON) (**c**), December–January–February (DJF) (**d**).

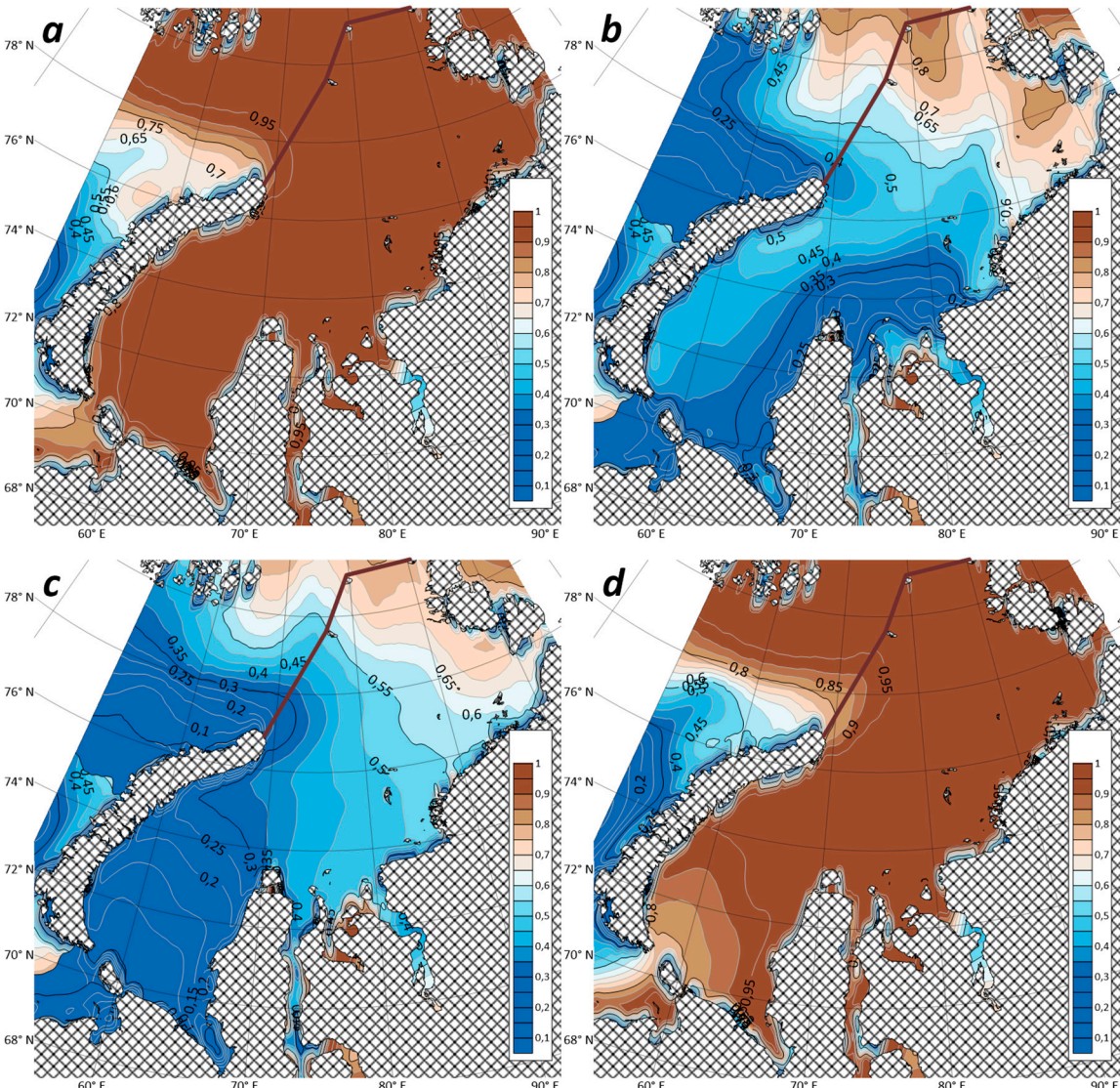

**Figure 9.** The probability of the presence of ice with a concentration of more than 50% in the Kara Sea according to reanalysis data (in 0–1 unit) for the periods: MAM (**a**), JJA (**b**), SON (**c**), DJF (**d**).

For the March–May period, the SWH does not exceed 4.5 m (Figure 8a) due to the ice presence for almost the entire period (Figure 9a). The Kara Sea is free from ice in this period for a short time and in small areas. There is only one local SWH maximum (8.1 m) in the southern part of the sea in June–August (Figure 8b). During this period, wind speed is usually less than in November–December therefore, severe storms are very rare despite the long ice-free period. Several SWH maxima are observed in September–November, including the 9.4 m height in the central part of the sea (Figure 8c). This maximum is an absolute multi-year SWH maximum for the central Kara Sea (Figure 5b). Ice occurs only in the northern Kara Sea in this period (Figure 9c). The strongest winds are observed in December–February. Most of the Kara Sea is ice-covered and the generation and propagation of wind waves are limited. However, severe storms from the Barents Sea pass to the northern Kara Sea during short ice-free periods. The absolute SWH maximum (9.9 m) for the entire sea was recorded there (Figure 8a). The differences in the wave characteristics in Figure 8a,d are mainly associated with the features of the atmospheric circulation because the ice distribution is very similar in March–May and December–February (Figure 9a,d).

### 3.3. Interannual Variability of Storm Wave Events

The number of storm events per year was calculated in the Kara Sea according to the POT method (the technique is described in Section 2.3). The events have different SWH thresholds from 3 to 7 m. Next, we will call these storm events with a different wave height simply as a storm. At first, we analyzed the number of storms for each year (Figure 10), which called the recurrence of a storm. Cases of storms with the SWH ≥ 3 m were observed about 30 times per year, with maxima in 2016. The number of storms with the SWH ≥ 4 m is about 15 times. The most severe storms with a SWH threshold 7 m were not registered each year. It is noteworthy, that in 2016 peaks were also registered for all other SWH thresholds, and the recurrence of the most severe events ≥7 m is the highest. A local maximum number of storms with SWH thresholds 3 and 4 m was noted in 1995. The minimum numbers of storms for several SWH thresholds were noted in 1998 and 2003. A linear positive trend in the number of storms is observed for almost all SWH thresholds. A double increase in storm recurrence was observed for cases with thresholds 3–5 m from 1979 to 2017. It is worth noting that there is high interannual variability in the number of storms. The average variance is about 25–30% from year to year.

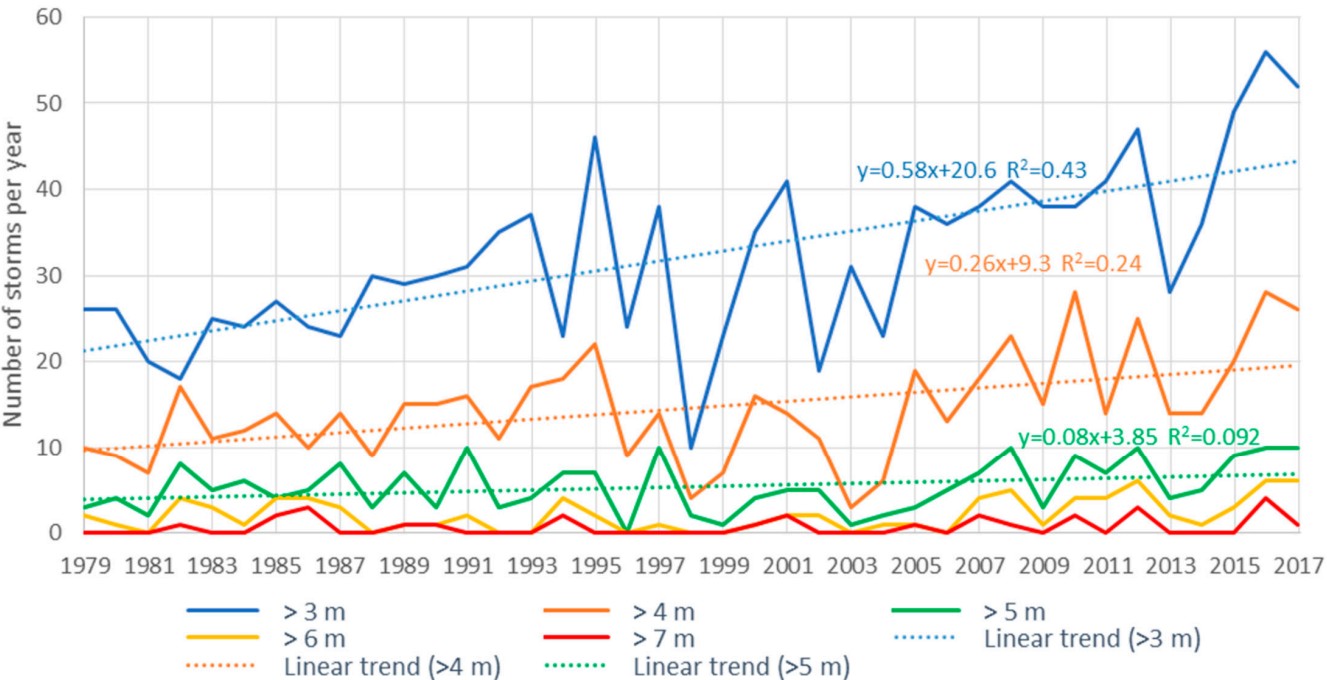

**Figure 10.** The number of storms with different SWH thresholds per year and its linear trends for 1979 to 2017.

The significance of trends was assessed by the F-test. The F-statistic is the standard significance test of the linear model. We have applied the F-test statistics of the analysis of variance (ANOVA) approach, which based on the null-hypothesis that the means of a given sample of normally distributed populations, all having the same standard deviation, are equal. Trends for the number of weaker storms more than 3–4 m are significant at the level $p = 0.05$. For more severe storms with SWH thresholds 5–7 m, trends are statistically insignificant. A similar result was obtained from 2005 to 2018 [19].

The analysis of the ice concentration variability in the Kara Sea was performed to explain the interannual variability of the storminess. The graphs of ice probability for two points in the Kara Sea (points location shown in Figure 6) are presented in Figure 11. Ice probability is the ratio of the number of days with observed ice to the duration of the whole year. The points were selected in the central and southern parts of the Kara Sea to demonstrate the difference of the ice conditions. There is a significant negative trend in the variability of ice period duration. Ice probability is approximately twice as less from 1979 to 2017. This trend is observed at both points therefore the ice period decreases in the most

part of the sea. This fact has been detected by various researchers previously [23,24,53–55]. The ice probability decreases from 0.7 to 0.55 in the center of the Kara Sea (T1 point). The decrease is even greater in the southern Kara (T2 point): From 0.7 to 0.4 (Figure 11). A local minimum of the ice probability was noted in 1995 in the southern part of the Kara Sea. This minimum probably led to an increase in the number of storms with SWH $\geq 3$ and 4 m (Figure 10). However, in 1995 ice cover reduction was observed only in the southern Kara Sea, not in the whole sea, that is why such reduction does not cause extreme storms ($\geq$5 m). The maximum ice cover was observed in 1998–1999 and amounted to 0.8 in both points. It led to the storminess weakening (Figure 10). The ice probability minima were observed in 2012 and 2016 in T1 and T2. These minima coincide with a significant increase in the number of storms (including storms with SWH $\geq 7$ m) that was observed exactly in these years.

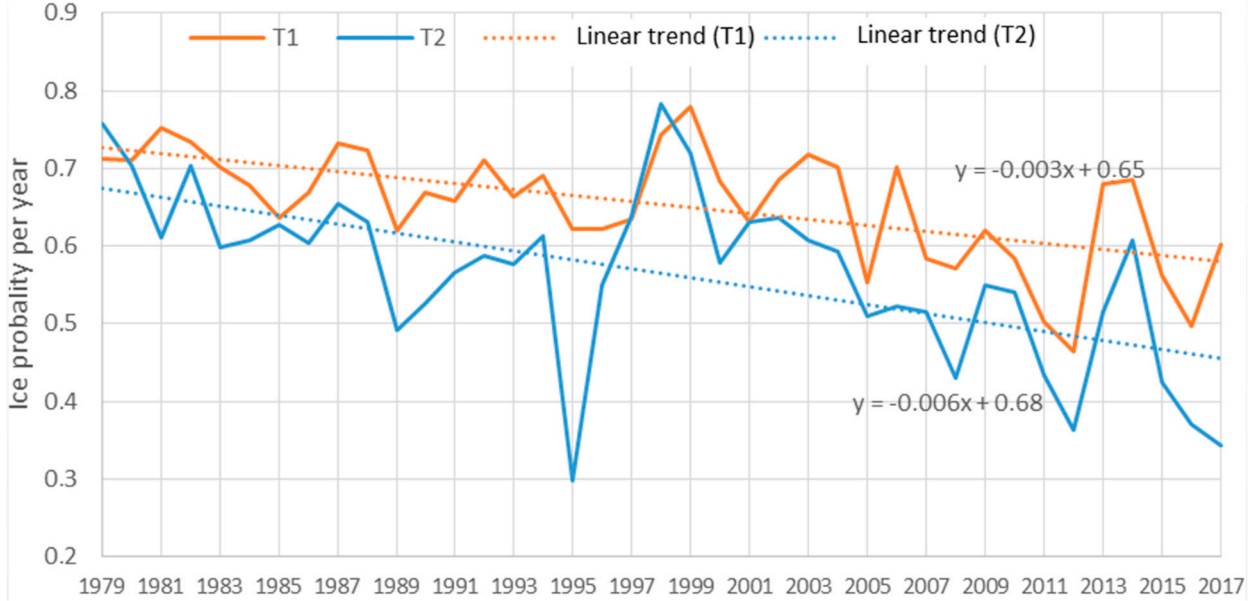

**Figure 11.** The probability of the ice presence with a concentration of more than 50% for two points in the Kara Sea by years.

Also, we analyzed the interannual variability of the wind conditions in the Kara Sea to explain the interannual variability of storm recurrence. The relationship between wind speed and wave height is non-linear. In addition, we need to consider such factors as fetch length, ice presence, and duration of wind impact. Therefore, correlation analysis for the wind recurrence with defined speed (higher than threshold values) and wind duration time with the storm repeatability was performed. The average daily wind speed at 10 m above the sea level was obtained from the reanalyses NCEP/CFSR and NCEP/CFSv2 for the period 1979–2017 for two points (the same as for the ice probability analysis points): T1, 66.04° E, 73.91° N; and T2, 61.59° E, 71.09° N (these points showed on Figure 6). The maximum correlation (0.65) is observed in a comparison of the number of storms with SWH threshold 4 m and wind recurrence with speeds greater than 10 m/s, it was revealed for two continuous days at T1. These storm wind conditions were used as an indicator in the analysis of the interannual variability of the wind.

The recurrence of storm wind conditions, the number of storms with SWH more than 4 m, and the ice probability are shown in Figure 12. The recurrence of storm wind conditions agrees quite well with the recurrence of storms. It is also seen that years with a long ice period reduce the number of storms in 1998 and 2003 despite the average values of storm wind conditions. The significance of trends was estimated by F-test. Trends of the storm waves recurrence and the ice probability are significant at the level of 0.05, and the trend of the storm wind recurrence is statistically insignificant.

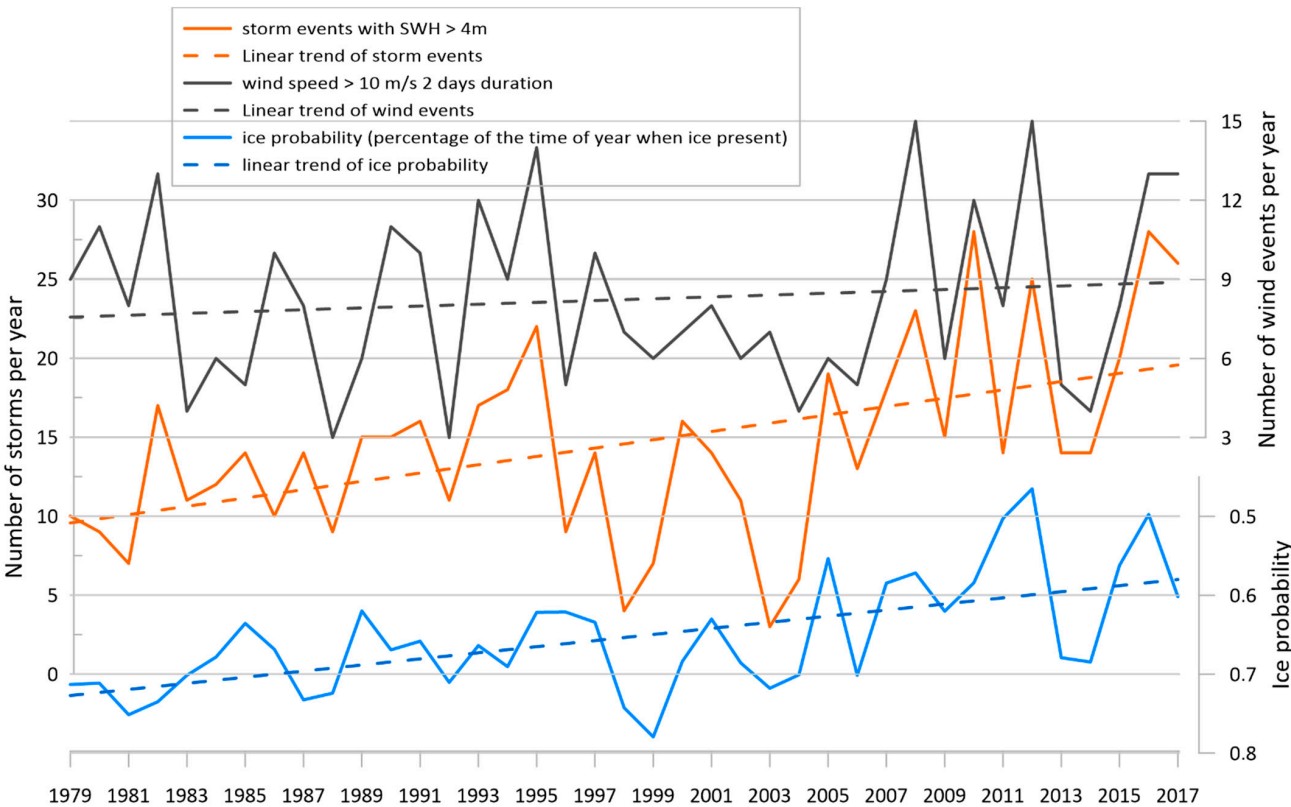

**Figure 12.** Recurrence of wind speed of more than 10 m/s and 2 consecutive days at T1 point, the number of storms with a SWH threshold 4 m, and probability of the ice presence at T1 point (opposite scale).

Thus, there is an evident positive trend for the number of storms in the Kara Sea according to the results of the analysis. This trend is mainly caused by the sea ice cover decrease over the past 40 years, the trend of storm wind conditions is not statistically significant. The interannual variability of events with SWH more than 3–4 m correlates quite well with the wind recurrence (speeds more than 10 m/s). However, both wind and ice conditions, certainly affect the storminess. Ice cover reduction leads to an increasing of weaker storms SWH $\geq$ 3–4 m in the southern Kara Sea. Such a reduction in the entire sea leads to an increase in extreme storms number (SWH > 5–7 m). The influence of ice cover variability also was obtained in the work [18].

Climate changes in storm wind conditions may be associated with changes in the ice conditions in the Kara Sea; however, this analysis is already beyond the scope of our research and requires a more detailed study. It is a challenge task for future research.

## 4. Probability Analysis of Storm Waves

### 4.1. Probability Functions of the Storm Recurrence in Different Sectors of the Kara Sea

Based on the analysis of the mean long-term and seasonal variability of the wave heights, the Kara Sea was divided into several sectors with different wave conditions (Figure 13). In these areas, several zones of maximum waves are observed in different periods of the year (Figure 8a–d). This segmentation allows us to analyze extreme storms in detail.

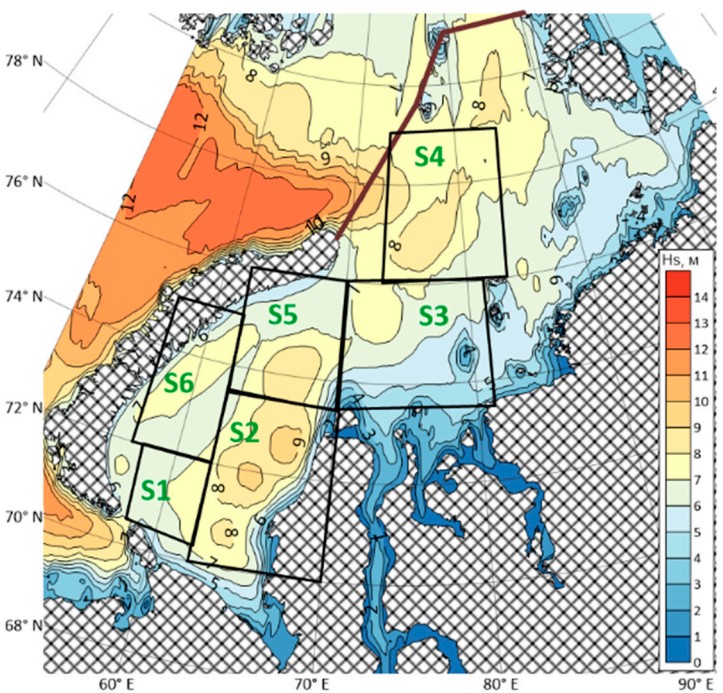

**Figure 13.** The SWH maximum and segmentation of the Kara Sea: Six sectors with different wave conditions.

A catalog of storms with SWH more than 3 m was formed for each sector shown in Figure 13. The POT method was used to create the catalog, and a SWH threshold of 3 m was chosen as the 95th percentile for the sample for the ice-free period. In this catalog, each member of the series is a separate storm event. It is a necessary condition for the independence of the members of the series according to the method of "independent storms" [56]. The length of the data series is sufficient for statistical analysis. Series consists of 450–750 values depending on the sector.

The storm data series for each of the six sectors were approximated by various distribution functions. We aimed to select the best distribution not only in terms of the best fit, but also in terms of a small number of parameters and to keep physical sense. Therefore, we did not consider any combined or complicated distribution laws (e.g., mixed laws). We used three distributions to fit the storm data series, including the Generalized Pareto, Gumbel, and Weibull distributions. Data fitting was analyzed for six sectors via qq-plots and cumulative distributions in logarithmic form. Statistical evaluation has shown that in all cases the best fit refers to the Pareto distribution, which is a special case of the Generalized Pareto distribution with scale parameter $H_{th} = \sigma / k$, and shape parameter $\gamma = 1 / k$. We tested the abovementioned distributions and got the following result: $R^2 = 0.67$ for the Gumbel distribution, $R^2 = 0.75$ for the Weibull distribution and $R^2 = 0.96$ for the Pareto distribution. A comparison of the functions with the empirical data showed that the best approximations for the storm recurrence was the Pareto distribution

$$F(H) = 1 - \left( \frac{H_{th}}{H} \right)^{\gamma} \tag{6}$$

where $H_{th}$ is the threshold value. $\gamma$ is the distribution parameter easily determined by the least square. For this purpose, formula (6) by logarithm is reducing to

$$ln(1 - F(H)) = -\gamma \cdot \ln(H) + \gamma \cdot \ln(H_{th}) \tag{7}$$

The transformed values of these parameters (according to Equation (7) in a logarithmic form) are presented in Figure 14 at each plot as linear equations coefficients. The

determination coefficients in all cases are more than 0.96, which confirms robust data fit, except for the most extreme values.

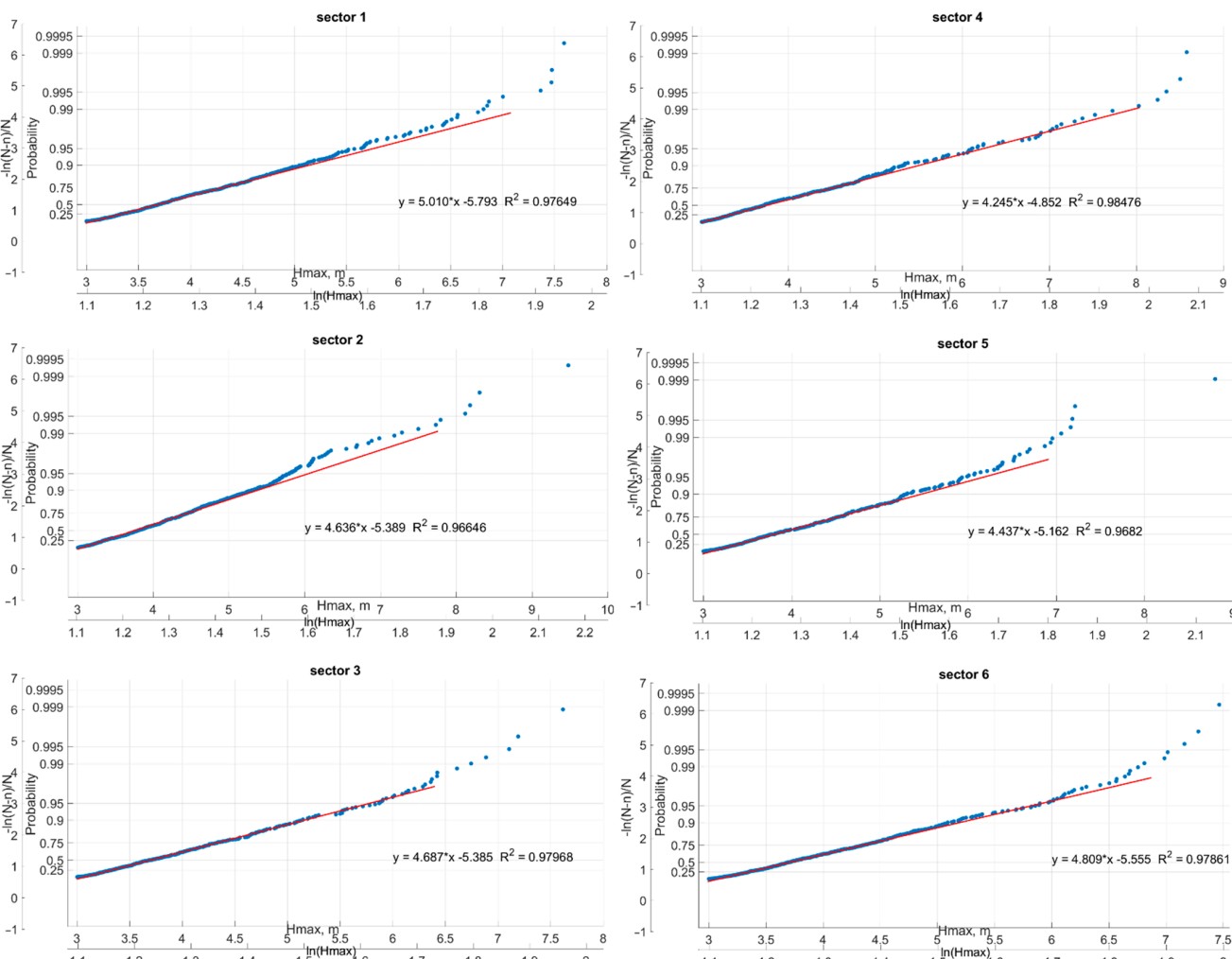

**Figure 14.** The empirical probability distribution of storms with different wave heights for each of the six sectors, presented in the Ppareto logarithmic coordinates. The determination coefficient of determination and regression equations are given for each sector.

If the empirical values on the diagram are located along a straight line in the logarithmic coordinates, this means that the empirical distribution fits well the Pareto distribution. The quantitative correspondence of the empirical and the theoretical distribution is established.

Pareto distribution of storms with different SWH for all sectors is shown in Figure 14. About 99% of the data fit by Pareto distribution (with confidence interval is 95%) with parameters $H_{th}$ = 3 m and $\gamma$ = 4.8 and a determination coefficient of $R^2 \approx 0.98$ in sector 6. This approximation is used as base distribution. A similar pattern of distribution functions is observed for all six sectors.

The average value of $\gamma$ is equal to 4.6 (varying from 4.2 to 5.0 for different sectors). The proximity of the parameters in the Pareto distribution indicates that the extremes are generated in all sectors with a similar law. Thus, the wave generation with an SWH more than 3 m is determined by the same mechanism. The basis of the hypothesis is the series of an extreme determined by the same law of probability distribution. We referred the most extreme events fitted the main sample distribution to the "black swans", and the main set of extremes referred to "swans", or "white swans". This indicates the "black swans" are the

most extreme events fitted this pdf. However, there are very rare cases when the empirical distribution deviates and exceeds the base distribution in the largest values area. Such patterns occurred in a broad range of science, originating from extreme events and values analysis in economics, demography, turbulence fluctuations, fires distribution, etc. [57]. These values, which do not obey the main distribution law, but do the other law with substantially different parameters, referred to "kings", or "dragons" [58], denoting that its nature is principally different from "black swans". We have successfully applied this terminology in the extreme wind speed analysis based on station and model data [59–61].

*4.2. Interannual Analysis of Extreme Events ("Dragons")*

In our case, several extreme values that deviate from the base distribution were detected in each sector due to the analysis of the distribution functions. The basic distribution ends in the range of SWH values 6.5–8 m in different sectors. The mentioned deviations have the common upward direction (Figure 14). Unique extrema "dragons" falling out of the base distribution, the most extreme values of SWH greater than 8 m were observed in 2, 4, and 5 sectors. They have a different distribution law and, probably, a different genesis.

It is critical that the probability of extreme events is based on a theoretical function, in our case, the Pareto distribution. For example, the data of sector 6 (East coast of Novaya Zemlya) (Figure 14) shows an SWH equal to 6.7 m (logarithm 1.9)—almost the last value that still lies on the base Pareto distribution. This value repeated through 47 sample elements $\left( \left( \frac{H}{H_{th}} \right)^{\gamma} \right)$ on average. However, storms with such SWH occurs about 100 times in reality (Figure 14), twice as much as it was planned by the Pareto approximation. A similar situation is reflected in the other sectors in the "dragons" zone. The use of base distribution in this zone leads to incorrect probability calculation results. This fundamental result demonstrates the source of systematic errors in evaluating the recurrence of extreme wave heights, which are especially relevant in applied and forecast tasks.

The probability of "dragons" does not match the base distribution. In the Kara Sea, the occurrence of storms with high waves depends on several factors simultaneously: Primarily on the wind speed, direction, and duration of the wind, secondly on the ice conditions (fetch limit) or the influence of the Barents Sea (for the 4-th sector). The number of storms with SWH more than 3–4 m is closely related to wind speed and wind duration as it was shown in Section 3.2, but the repeatability of storms with SWH more than 6–8 m requires the simultaneous combination of small ice cover and extreme wind conditions. Thus, the division of the empirical distribution function between "black swans" and "dragons" occurs when the influence of small ice cover (and consequently longer fetch) is observed besides the wind forcing. Since wind and ice conditions are considered as approximately independent events, their joint probability is much lower than the probability of rare wind events.

Extreme events with any (even very huge) wave height can occur according to the base distribution function, formally. However, the empirical function for the "dragons" is nonlinear and goes to a certain plateau; it was shown in the logarithmic graphs. The specific distribution parameters for "dragons" samples showed there are significant differences in fitting. Sectors 2, 3, and 6 demonstrated quite linear approximation more like the main "swans" sample. It is important to highlight that most "dragons" samples fit well to other parameters of the Pareto distribution in linear approximation ($R^2$ is more than 0.9 and significant in most cases), which indicates they could obey the other law and physical processes. However, other sectors showed strong deviations of "dragons" approximation parameters from the "swans" ones. Exceptions are marked outliers (sectors 2 and 5), which do not fit any version of Pareto distribution. The $\gamma$ values (starting with some values of H) begin to increase rapidly. Particularly, the marked outliers in sectors 2 and 5 corresponds simultaneously to absolute maxima in the Kara Sea (above 9 m) and does not fit to the "dragons" sample. Thus, the "dragons" pattern could indicate a certain natural limit observed for maxima wave height that differs from the base distribution. When several cases from the data set do not match the base distribution, it could indicate a chaotic

behavior of the most extreme waves in these sectors, and this pattern has some similarities with the definition of freak waves in the article [62]. Freak waves are unique anomalous individual waves that do not correspond to the general distribution. In our case, we have a similar picture on the synoptic scale, where specific storms with a certain SWH maximum defined as "dragons".

Figure 15 depicts the "dragon" cases frequency (number of cases) within the 1979–2017 period for each of the six sectors. This frequency is observed to increase, especially after year 1998, it occurred in sectors 1 and 5 after 1997–2000 only, particularly. Comparing this tendency of the "dragon" cases and the sea ice cover trends over the Kara Sea (Figure 12), a higher frequency of "dragons" indicated simultaneously with peaks of wind recurrence and small sea ice cover (e.g., the years 2010, 2012, 2015).

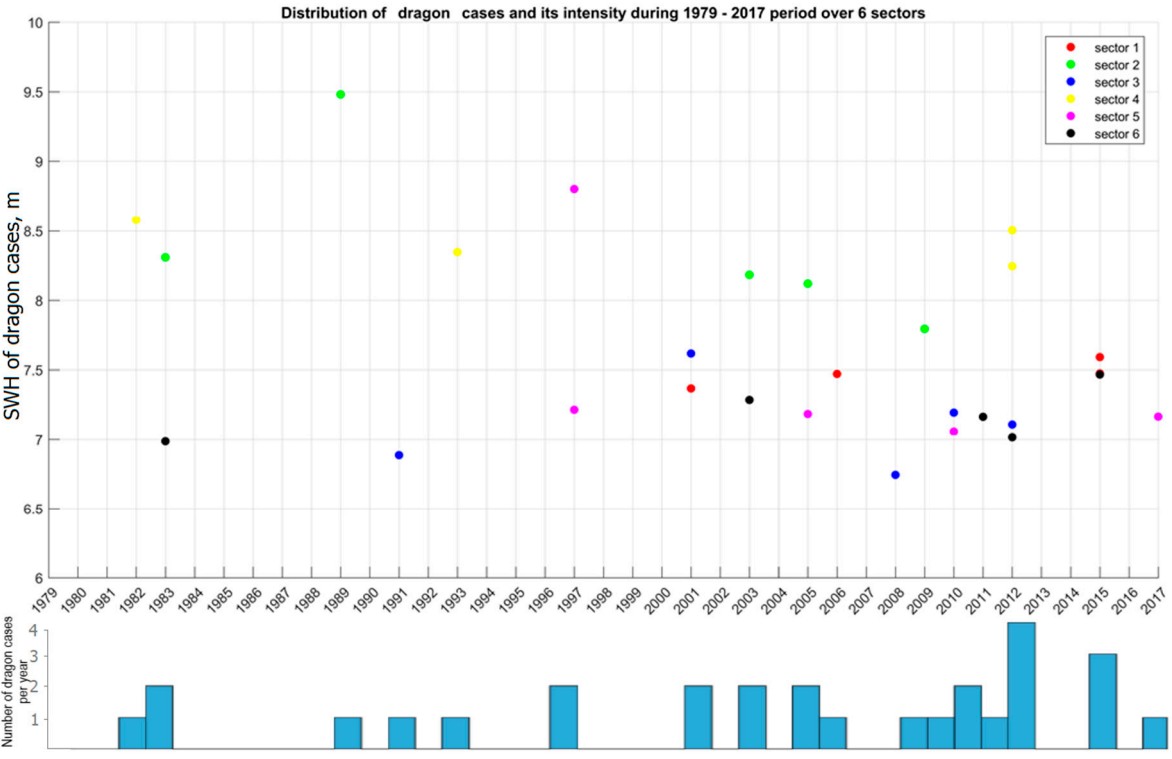

**Figure 15.** Cases of extreme events ("dragons") from 1979 to 2017 for all sectors of the Kara Sea.

Thus, a reduction of the sea ice cover (Figure 11) and increased recurrence of stronger winds (Figure 12) lead to an increase of the extreme wave height frequency (Figure 15), which is consistent with [18]. Therefore, the possible reason for statistically significant increase of the "dragon" SWH repeatability could be the ice cover reduction in the last 40 years.

## 5. Discussion and Conclusions

This article presents new information about wave climate and storm recurrence in the Kara Sea based on the results of wave modeling. The SWH, the mean WP, and mean WL fields were obtained for every three hours from 1979 to 2017 (39 years in total). The mean SWH for the entire sea varies from 1.1 to 1.3 m. The SWH maximum is 9.9 m and it was observed in the northern part of the Kara Sea. Analysis of the SWH maxima for different seasons showed that the SWH does not exceed 4.5 m in March–May. The wave generation is limited by the ice presence in some periods of the year. The long-term mean wave period value is 3.5 sec and the average wavelength is 30 m for the central Kara and 35–40 m for the northern part.

The storm recurrence with the SWH thresholds from 3 to 7 m was calculated in the Kara Sea for each year according to the POT method. Storms with the SWH $\geq$ 3 m are observed about 30 times per year on average, with the SWH more than 4 m–about 15 times. Storms with the SWH threshold of 7 m are rare and observed not every year. The storminess was higher in 1994–1995 and after 2008. The minimum numbers of storms were registered in 1998 and 2003.

The combined analysis of the storm activity, the recurrence of strong winds, and the ice probability was conducted. The high recurrence of strong winds and the absence of sea ice lead to storm numbers increasing (SWH 3–4 m) in the southern Kara Sea. When the sea ice probability decreases for the whole sea and the recurrence of strong winds is high simultaneously, then the number of extreme storms (SWH more than 5–7 m) increases.

The linear trend of the storm activity is positive in the Kara Sea. A double increase of storm recurrence was observed for cases with SWH thresholds 3–5 m from 1979 to 2017. The linear trend of the severe storm recurrence (SWH more than 5–7 m) is positive but statistically insignificant because such events are rare. This trend is mainly caused by a reduced sea ice cover over the past 39 years, because the trend in recurrence of storm wind conditions is not significant.

Since the main results are based on wave modeling, can they be considered reliable? We get the BIAS 0.07–0.09 m, the RMSE is 0.31–0.39 m, and the errors are growing for SWH more 2–3 m. These errors could be critical for estimates of such parameters as the distribution of the long-term maximum SWH. Our study is reasonable for the mean long-term wave parameters as BIAS is near zero. The storm recurrence errors mentioned above could lead to an increase or decrease in the number of storm events with different SWH thresholds, but the errors affect neither the interannual variability nor the climatic trends of the storm recurrence.

The quality assessment of the wave model for extremely high waves should be explored. Unfortunately, the authors do not have full-scale direct measurement data in the Kara Sea, and satellite data also limited by the SWH 6–6.5 m.

Satellite data analysis is another way to study the storm activity, which is described in several articles [4,5]. A research period was limited to 15–20 years in these articles, moreover, not all satellite data has a good quality and should be verified [63,64].

This study corresponds to the storm activity results of Duan et al. [19]. There were estimates of the linear trends of the mean SWH and the extreme SWH from 2005 to 2018. Interannual variations of the mean SWH and the extreme SWH in the central part of the Kara Sea are shown in Figure 16. Duan et al. studied extreme SWH in several points as the 90th and 99th percentiles for each year. The minimal extreme SWH was in 2013 and it was growing since 2014. The same pattern was discovered in our results (Figure 10). However, the other events in 2007–2008 are not confirmed by our results. The "percentile" method for extreme wave analysis differs significantly from the POT method, but the strongest climatic trends can be identified. The trends in [19] are not statistically significant, as they are represented by simple statistical percentiles (with no separate storm events analysis), and the analyzed period was too short. Our results broaden the understanding of the storm activity in the last 39 years.

We also obtained the results of interannual storm wind variability. Trends of the recurrence of storm wind conditions are statistically insignificant (Figure 12). These results correspond to the studies [22,31]. However, the methodological approaches were different, so the comparison was general. The recurrence of storm waves depends not only on the wind speed but also on wind direction. If the wind direction changes in climatic terms, it could affect the wave growth (in the case of limited fetch). The climatic changes of the wind direction and its influence on wave heights require additional research.

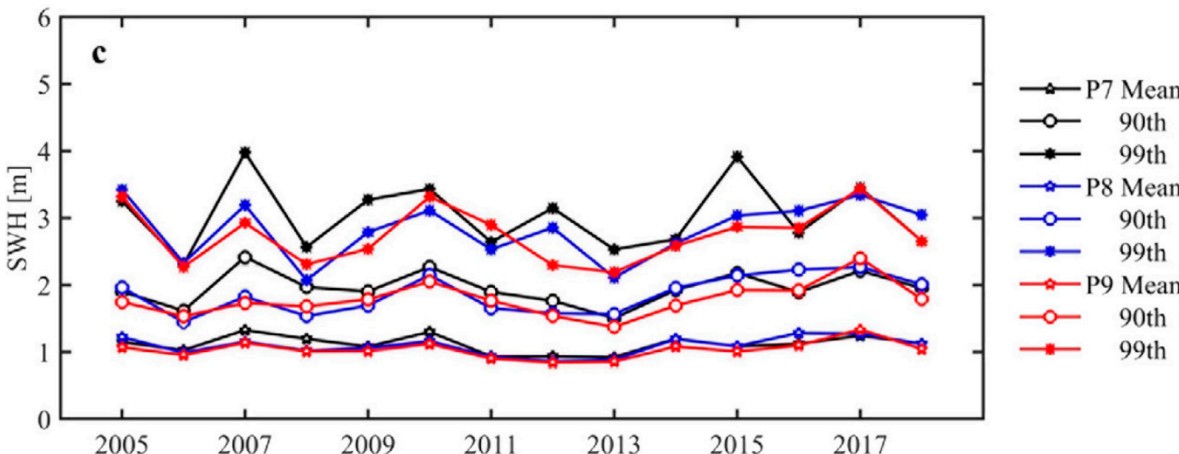

**Figure 16.** Interannual variations of mean and extreme SWH in the Kara Sea (Duan et al., 2019).

One of the crucial results of this study is the probability analysis of the storm events in different sectors of the Kara Sea. This laborious and exceptional part of the analysis is performed during a wave climate research rarely. The Kara Sea was divided into six sectors with different wave conditions due to the mean long-term SWH and seasonal variability of the wave heights.

Different approximations of storm events distribution were compared with the empirical distribution. The statistical evaluations showed Pareto was the best among others and satisfied the criteria of the best fit, small number of parameters and maintenance of physical sense. The proximity of the parameters in the Pareto distribution indicates that the extreme generation occurs in the same way for all sectors of the sea.

Analysis of the storm distribution functions for each sector showed that several extreme events ("dragons") deviate upward from the base Pareto distribution. Thus, the division of the empirical distribution function into "black swans" and "dragons" occurs when the influence of small ice cover (and consequently longer fetch) is observed besides the wind forcing. "Dragons" occurred in sectors 1 and 5 only after 2000, when the increased recurrence was registered for the entire Kara Sea. A higher recurrence of "dragons" was registered in years when there were simultaneous peaks of wind recurrence and small sea ice cover. On a timescale of 39 years, climatic changes indicated in increasing the recurrence of such extreme events as "dragons".

Our results are relevant for the climate variability studies and atmosphere-ocean turbulent heat fluxes interactions. In the case of the storm, wind waves could increase the turbulent heat fluxes significantly [65]. On the other hand, increased heat flux to the atmosphere could lead to the fast recovery of the ice in winter season, which will lead back to a decrease in the storm activity. Such negative feedback could exist in the climate system and could be a new direction for future research.

In this paper, we do not consider the relationship between the storm number variability and global climatic indices of the large-scale atmospheric circulation. Earlier in [46], we showed that the number of storms with SWH $\geq$ 7 m (DJF period) has a low correlation with the Arctic Oscillations index in the Barents Sea. Such correlation is caused mainly by the decisive influence of the Atlantic on the Barents Sea. In the Kara Sea, the influence of the Atlantic and Western transport is even less. Thus, there could be no connection with Arctic Oscillation here probably. This study shows that the wave climate in the Kara Sea is regulated by ice cover variability. However, there was a correlation between sea ice loss and the Arctic Oscillation detected in [66]. The connection between the wave climate of the Kara Sea and global indexes, what are we going to do in the future.

**Author Contributions:** The concept of the study was jointly developed by S.M. S.M. did numerical simulations, analysis, visualization, and manuscript writing. V.P. and A.K. did the probability analysis of storm waves and their visualization. K.S. analyzed the results of numerical modeling. I.M. validated the model. S.M. prepared the paper with contributions from V.P., A.K., K.S., and I.M. All authors have read and agreed to the published version of the manuscript.

**Funding:** The wave modeling and probabilistic analysis were done with the financial support of the RFBR (project 18-05-60147 Myslenkov S.A., Platonov V.S., Kislov A.V.). Data analysis funded by the Ministry of Science and Higher Education of Russia, theme 0128-2021-0002, and RFBR project 20-35-70039 (Silvestrova K.P). Validation of the model done by I.P. Medvedev with the financial support of the RFBR (project 18-05-60250).

**Acknowledgments:** Authors gratefully thank A.Yu. Medvedeva for the editorial remarks.

**Conflicts of Interest:** The authors declare no conflict of interest.

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
