# Peer review of "Thirty-Nine-Year Wave Hindcast, Storm Activity, and Probability Analysis of Storm Waves in the Kara Sea, Russia"

_water, doi:10.3390/w13050648_

Round 1

Reviewer 1 Report

The manuscript shows the regular and extreme wave characteristics in the Kara Sea by long-term wind wave modelling with WaveWatch III 4.18 and 6.07.

It consists of mainly two parts; 1) wind wave modelling for 39 years and the resulting wave characteristics and 2) extreme wave analysis with Pareto distributions with storm waves sector-by-sector in the Kara Sea.

It seems that this manuscript is a part of series such that some information on model build-up and extreme value analysis are just simply described.

The overall organization of Sections is better to be revised in more precise way. For example, make a material section in Chapter 2 for illustrating all the datasets that are used for the wind wave modelling from the bathymetry, surface wind forcing, reanalysis data, and so on. Also each section in Results are quite long so that it would be better to divide Results with more sections.

Therefore, it needs a major revision with a minimum improvement to the following comments.

General comments:

  1. It seems that the model output has 3 hour interval. It seems too long to discuss extreme waves from the 3 hour interval output. It needs more explanation or justification on the extremes that can not be captured with 3 hour interval.
  2. The information on the surface wind forcing is not enough. Should be describe on what and how the surface wind forcing in the wind wave modelling is obtained and prepared.
  3. SI (scatter index) should be defined in the text. and What is the definition of scatter index? It should be described clearly in the text.
  4. In line 235 - 236, "How ... absolute errors." >> needs a reference.
  5. In Figure 3(b), R, Bias, and RMSE should be provided.
  6. From line 288, it is better to divide into new section such as "seasonal wave characteristics".
  7. In Figure 10, the label for green plain line should be > 5m.
  8. From line 386, it should be a new section "Interannual variability"
  9. Section 3.3 is better to be a new chapter 4.
  10. From line 483 to line 522, in the extreme wave analysis, it is described that the Pareto distribution can not explain the extreme waves in the "dragon" zones. And, authors mentioned the distribution of the extreme waves in the dragon zones has some similarities with the definition of freak waves. Then, I wonder why the authors do not make the extreme wave analysis into two parts; "swan" and "black swan" with the Pareto distribution and the "dragon" with other extreme value analysis methods.
  11. Therefore, the authors should make additional analysis for the "dragon" zones with other extreme value analysis methods to explain its mechanism.
  12. The terms such as "swan" and "black swan" and "dragon" are quite unfamiliar with, at least, me in the wind wave modelling. If the authors want to use these terms, then they should be clearly defined in the text with clear classification criteria. 
  13. There are many ambiguous terms as well in the text. Please, revise the whole text carefully in terms of definitions of technical terms.
  14.  For line 518 - 519, "This graph was analyzed for ... reccurrence." needs more evidence or a reference.
  15. For line 525-527, it needs to be proven or a reference. For example, combine Figure 15 and Figure 11.
  16. Line 554, epy storm wind?
  17. Figure 16 needs more clear explanation. and What is the definition of the interannual variations exactly?
  18. Line 588, the best approximation was the Pareto distribution? How about the "dragon" zones? This paper is now discussing the extreme wave characteristics, right? I think the "dragon" zones are the real extremes which are not fitted with the Pareto. Then, it is not the best. I think the authors can do further.

Reviewer 2 Report

The paper aims to assess the impact of climate change on storm activity over a period of 39 years in the Kara Sea, which is of the interest of the journal’s audience. However, there are also concerns in the current version of the manuscript.

  1. The title of the paper is not consistent with the main objective of the study. It should be modified, and “ the Kara Sea, Russia” can be used.
  2. In the introduction, the literature is not up to date. The latest long term analysis and extreme value analysis techniques are not cited.
  3. Section 2.1 is a standard model and has little new contribution.
  4. Section 2.2, POT technique should be one of the key method in the analysis, but was not introduced clearly. For POT, SWH = 3 m is selected, what is the justification and how sensitive it is? How were results validated?
  5. Some of the presented figures need to be improved. For example, Fig 4 axis label font size is too small RMSE unit? Fig 13 what are the color contour. Figure 14 is not clearly presented, R2, significant number in figure and text are not consistent. Figure 16 is in poor quality.

A major revision is recommended.

Round 2

Reviewer 1 Report

The authors make appropriate responses to all comments.

One minor point.

Section 4. Discussion and conclusion should be Section 5.

Author Response

We fixed this mistake.

Thank you!